# EEG-Based Multi-Modal Emotion Recognition using Bag of Deep Features: An Optimal Feature Selection Approach

**DOI:** 10.3390/s19235218

**Published:** 2019-11-28

**Authors:** Muhammad Adeel Asghar, Muhammad Jamil Khan, Yasar Amin, Muhammad Rizwan, MuhibUr Rahman, Salman Badnava, Seyed Sajad Mirjavadi

**Affiliations:** 1Department of Telecommunication Engineering, University of Engineering and Technology, Taxila 47050, Pakistan; adeel.asghar@students.uettaxila.edu.pk (M.A.A.); muhammad.jamil@uettaxila.edu.pk (M.J.K.); engr.fawad@students.uettaxila.edu.pk (F.); yasar.amin@uettaxila.edu.pk (Y.A.); 2Department of Computer Engineering, University of Engineering and Technology, Taxila 47050, Pakistan; muhammad.rizwan@uettaxila.edu.pk; 3Department of Electrical Engineering, Polytechnique Montreal, Montreal, QC H3T 1J4, Canada; 4Department of Computer Science and Engineering, College of Engineering, Qatar University, P.O. Box 2713 Doha, Qatar; 5Department of Mechanical and Industrial Engineering, College of Engineering, Qatar University, P.O. Box 2713 Doha, Qatar; seyedsajadmirjavadi@gmail.com

**Keywords:** emotion recognition, brain computer interface, bag of deep features, continuous wavelet transform

## Abstract

Much attention has been paid to the recognition of human emotions with the help of electroencephalogram (EEG) signals based on machine learning technology. Recognizing emotions is a challenging task due to the non-linear property of the EEG signal. This paper presents an advanced signal processing method using the deep neural network (DNN) for emotion recognition based on EEG signals. The spectral and temporal components of the raw EEG signal are first retained in the 2D Spectrogram before the extraction of features. The pre-trained AlexNet model is used to extract the raw features from the 2D Spectrogram for each channel. To reduce the feature dimensionality, spatial, and temporal based, bag of deep features (BoDF) model is proposed. A series of vocabularies consisting of 10 cluster centers of each class is calculated using the k-means cluster algorithm. Lastly, the emotion of each subject is represented using the histogram of the vocabulary set collected from the raw-feature of a single channel. Features extracted from the proposed BoDF model have considerably smaller dimensions. The proposed model achieves better classification accuracy compared to the recently reported work when validated on SJTU SEED and DEAP data sets. For optimal classification performance, we use a support vector machine (SVM) and k-nearest neighbor (k-NN) to classify the extracted features for the different emotional states of the two data sets. The BoDF model achieves 93.8% accuracy in the SEED data set and 77.4% accuracy in the DEAP data set, which is more accurate compared to other state-of-the-art methods of human emotion recognition.

## 1. Introduction

Brain–computer interface has been used for decades in the biomedical engineering field to control devices using brain signals [1]. The electroencephalogram (EEG) signals captured from the electrodes placed on the human skull [2] are used to classify and detect human emotions. Many researchers have conducted a lot of studies about the recognition of emotions through EEG signals. However, emotion recognition is still a challenging task for machines to recognize. With the advancements of machine learning tools, there is a growing need for automatic human emotion recognition [3]. Human emotional states are associated with the participant’s perception and apprehension. Emotional awareness is of great importance in other areas such as cognitive sciences, computer science, psychology, life sciences, and artificial intelligence [4]. Due to the growing demands of mobile applications, emotion recognition is also becoming an essential part of providing emotional care to people. Human emotions can be recognized from speech, image, or video graphics, but the system for these types of recognition systems are much expensive. The task of recognizing emotions from brain signals is yet challenging due to the lack of temporal boundaries; also, different participants perceive the unusual amount of emotions in different ways [5]. Previously researchers have found new ways to understand and discover emotions through speech, images, videos, or BCI technology. In connection with non-invasive techniques, the brain–computer interface (BCI) provides a gateway for obtaining EEG signals related to emotional stimuli. The signals collected by the BCI help to better understand the emotional response, but it is still unclear how we can accurately and extensively decipher emotions [6,7]. BCI and biosignal acquisition techniques have grown considerably allowing real-time analysis of biosignals to quantify relevant insights such as the mental and emotional state of the user [8]. EEG signals are collected using 10–20 international systems for electrodes placement used to decode the information [9,10].

Despite many of BCI’s techniques for capturing EEG signals for emotion recognition, however, there is still room for improvement in extracting spatial functions, including accuracy, interpretability, and utility of online applications. The most consistent approach proposed by [11] is to normalize the common spatial pattern (CSP). Normalized CSP is used to extract features to achieve good decoding accuracy. The purpose of normalizing the CSP is to reduce the effects of noise and artifacts that occur in the raw EEG signal. Ref. [12] also used CSP features for motor rehabilitation in virtual reality (VR) control.

Ref. [13] designed a filter for selecting features using CSP. As experienced by previous BCI researchers [13,14], the manual selection of the best filter for each subject is still tricky. Ref. [13] suggested optimal filtering to select filters for all subjects automatically. Choosing the best filter removes the access noise from the signal, resulting in a superior accuracy of the classification performance.

In the past, researchers used time-frequency distribution and spectral analysis methods, such as the discrete wavelet transformation method (DWT) [15] and the Fourier transformation method (FT) [5]. However, given the complex and subjective nature of the emotional state, it is challenging to introduce a general method for analyzing different emotional states. For non-stationary EEG signals, the frequency components change with time and frequency component information is not enough for the classification of human emotions. Therefore, to acquire the full knowledge of signal frequency in the spatial and temporal domain, another technique used is a continuous wavelet transform (CWT). Several techniques for automated classification of Human emotions from EEG signals are proposed using different machine learning techniques. Due to the non-linear behavior of EEG signals recognizing emotions for a different subject is a challenging task. Therefore, the selection of channels and features is crucial in recognizing human emotions accurately. Large feature dimensions have high calculation costs and a broad set of training data. Various techniques proposed by the researchers [14,16] decompose the signal into a series of features to deal with extensive data. Wavelet-based techniques such as empirical mode decomposition (EMD), discrete wavelet transform (DWT), and wavelet packet decomposition (WPD) are used to decompose signals. Ref. [16] used multi-scale PCA along with WPD to decompose and remove noise from the signal. When classifying EEG signals for motor rehabilitation, they achieve a classification accuracy of 92.8%. The wavelets based feature extraction technique is proposed in [17] for emotion classification on the SEED dataset. They used flexible analytical wavelet transform (FAWT) for channel decomposition. FAWT decomposition is a channel-specific technique that selects specific channels based on machine learning. The accuracy of 83.3% is achieved using SVM on the SEED dataset. Ref. [18] proposed a method of evolutionary feature selection in which frontal ad occipital channels were selected for classification and achieved an accuracy of 90%. Feature selection methods are effective in eliminating irrelevant features and maximizing the performance of the classifier to reduce high dimensions automatically. Of the many techniques that can be applied to prevent selection problems, the simplest is a filtering method based on ranking techniques. The filter method selects functions by scoring and ordering tasks based on their relevance and defining thresholds for filtering out irrelevant features. This method is intended to filter less relevant and noisy features from the set of features to improve classification performance. Filtering methods applied to the emotion classification system include Pearson correlation [19], correlation-based feature reduction [20,21], and canonical correlation analysis (CCA). However, the filter method has two possible disadvantages, assuming that all features are independent of each other [22]. The first disadvantage is that there is a risk that features are thrown away that are not relevant when viewed separately, but that may be relevant in combination with other features. The second disadvantage is the ability to choose individual related functions that can cause duplication. To overcome this issue, the evolutionary-based feature selection method proposed by [23] evaluated on DEAP and MAHNOB dataset. Differential evolutionary (DE) based features selection method was classified using a probabilistic neural network (PNN) and achieved a classification accuracy of 77.8% and 79.3% on MAHNOB and DEAP datasets, respectively. Text-based and speech-based emotions are also proposed by [3], which used the Mel frequency cepstral coefficients (MFCC) and reported the overall accuracy of 71.04% on IEMOCAP dataset. Multivariate empirical mode decomposition (MEMD) in [15] also used to decompose channels up to 18 out of 32. It decomposes the signal into amplitude and frequency modulated (AM–FM) oscillations known as intrinsic mode features (IMFs). Two-dimensional emotional states in arousal and valence dimensions are classified in [15] using SVM and ANN classifiers. In other studies, differential entropy is calculated on different frequency bands associated with EEG rhythms. Beta and gamma rhythms are the most effective for emotion recognition [24,25]. The authors discovered that 18 different linear and non-linear features were time-frequency domain features using spatial-temporal recurrent neural networks (STRNN). Spatial and temporal dependency model is designed to select features. Ref. [26] investigated the dynamic system features of EEG measurements and other aspects that are important for cross-target emotion recognition (e.g., databases for different EEG channels and emotion analysis). The recursive emotion feature elimination (RFE) method proposed by authors to eliminate repetitive features ad reduce the feature dimension. They achieve an average accuracy of the rating of 59.06% and 83.33% Using physiological signals (DEAP) and SJTU sentiment EEG data set (SEED) databases, respectively. Recently, many articles have been published in the field of emotion recognition [27,28,29] with the help of EEG signals. In comparison with traditional methods that use deep learning, there is a possibility of recognizing emotions in multi-channel EEG signals. However, two challenges remain. First, is deciding how to obtain relevant information from the time domain, the frequency domain, and the time–frequency characteristics to the emotional state EEG signal.

Ref. [30] states that the selection of specific EEG channels is essential for multi-channel EEG-based emotion recognition. Ref. [30] shows 32 channels and ten specific channels (F3, F4, Fp1, Fp2, P3, P4, T7, T8, O1, and O2 for emotion recognition). Experiments show better results when 10 channels are used compared to all 32 channels. Ref. [31] suggested the emotion recognition method based on the entropy of samples. Their experimental results corresponding to channels related to the emotional state are primarily from the frontal lobe areas, namely F3, CP5, FP2, FZ, and FC2. Few studies have analyzed the spatial domain characteristics of multi-channel EEG, and it may also contain important information. There were also some spatial features. The study is limited to the asymmetry between electrode pairs [32].

In the past, researches used to recognize human emotion from the selective number of channels, which may increase the computational speed but decreases the accuracy rate of understanding emotions. In this paper, we presented an accurate multi-modal EEG-based human recognition using a bag of deep features (BoDF), which ultimately reduces the size of features from all the channels which are used for detecting brain signals. First, a feature vector is obtained from a time-frequency representation of a preprocessed EEG emotions dataset using continuous wavelet transform (CWT). The features of all subjects from the AlexNet model are collected, which uses 2D images as input. BoDF method is proposed to arrange features as a single matrix and reduce the feature vector using k-means clustering. The reduced feature vector for all channels is then classified. Three states of participants as positive, negative, and neutral for SEED dataset and two states as arousal and valence are classified using all kernels of support vector machine (SVM) and k-nearest neighbor (k-NN) classifiers.

The rest of this article is organized as follows: Section 2 describes the dataset and electrode channel mapping. Section 3 introduces the emotion recognition framework and the proposed deep feature model using SVM and k-NN classifiers. In Section 4, experimental results are discussed by comparing other emotion recognition models. Section 5 concludes this work.

## 2. Materials

### 2.1. SEED Dataset

The emotion recognition task is carried on SEED [33] dataset developed by SJTU. The EEG dataset was collected by Prof. Bao Liang Lu at brain-like computing and machine Intelligence (BCMI) laboratory. This publicly available dataset contains multiple physiological signals with emotion evaluation, which makes it a well-formed multi-modal dataset for emotion recognition. In this data set, 15 participants were subjected to watch four minutes of six video clips. Each clip is well-edited that can be understood without explanation and exhibits the maximum emotional meanings. The detail of each clip used in acquiring EEG data are listed in Table 1.

The data was collected from 15 Chinese participants (7 males, 8 females), who were aged between 22–24. Each participant’s data includes 15 trials, and, in each trial, the experiment performed twice. The RAW EEG signals were first segmented and then downsampled to 200 Hz. A bandpass frequency filter of 0–75 Hz was also applied to remove noise from the signal and EMG signals. The data was collected from 62 channels of each participant using 10–20 International standard [33].

### 2.2. DEAP Dataset

DEAP dataset is also publicly available as an on-line dataset for emotion classification [34]. The dataset was generated by a team of researchers at the Queen Mary University of London. The DEAP dataset contains multiple physiological signals for the evaluation of emotions. 32 channel EEG data were collected from 32 subjects. EEG signals were recorded by showing 40 preselected music videos, each with a duration of 60 seconds. The signals were then downsampled to 128 Hz and de-noised using bandpass and lowpass frequency filters. DEAP dataset can be classified using Russell’s circumplex model [35] as shown in Table 2.

To visualize the scale by Russell’s model DEAP dataset uses self-assessment manikins (SAMs) [36] with the real numbers 1–9. To make SAM as a continuous scale, each subject was asked to tick any number in between 0–10 from the list provided [34]. The scale based on self assessment rating were selected as 1–5 and 5–9 [37,38,39,40,41]. If the rating was greater than or equal to 5, the label was set to “high”, and if it was less than 5, the label was set to “low”. Thus, to create a total of four labels: high arousal low valence (HALV), low arousal high valence (LAHV), high arousal high valence (HAHV), and low arousal low valence (LALV). The emotional states describe by the given labels are shown in Table 2.

In this work, both the dataset were analyzed separately. In the SEED dataset, three classes are used as positive/negative/neutral. Positive class indicates the subject is in a happy mood; the negative class indicates the sad emotion of the subject while the neutral class tells the normal behavior of the subject. In the DEAP dataset, there are two types of labels named valance and arousal with a scale of 1–9. Labels 1–4 represent the valance, and 5–9 represents the arousal scale. Detailed information about the datasets can be found in [34].

### 2.3. Electrode to Channel Mapping

Both the datasets acquired EEG signals using the 10–20 International System [42]. The 10–20 international system describes the position of the electrodes to be placed on the human scalp for detecting EEG signals. The “10” and “20” indicate that the actual distance between adjacent electrodes is 10% or 20% of the distance between the front and back or right and left sides of the skull. Figure 1 shows the mapping of electrodes to channel number describes by both the datasets. The first 32 channels are used by the DEAP dataset, while all 62 channels are used by the SEED dataset to acquire EEG signals. In the figure, the first number in Figure 1 represents the channel number of the DEAP data set, and the second number is the channel number of the SEED data set (DEAP channel number, SEED channel number). Nasion in the figure represents the frontal part of the head, and channels are named using the 10–20 international system.

## 3. Methodology

This section explores step by step working of the proposed model and overall architecture. Figure 2 presents the framework of our work. First, the preprocessed EEG signals of both the datasets are used for time-frequency representation using the continuous wavelet transform filterbank. The feature extraction stage extracts the feature from all the channels of both the datasets.

### 3.1. Time Frequency Representation

The preprocessed EEG signal is used for time-frequency representation (TFR) of signals. Traditional emotion recognition models directly extract features from the EEG signals results in low accuracy and abortive results. In this paper, the one-dimensional EEG signal is presented as a two-dimensional model of EEG signal as time–frequency representation to analyze the signal in a better way and extract the desired features. This is achieved by using a continuous wavelet transform.

#### Contineous Wavelet Transform

Continuous wavelet transform (CWT) expresses the signals in terms of wavelet functions [43], which are localized in both time and frequency domain. CWT provides the complete representation of the 1D signal by letting the translation and scale parameters of the wavelets, varying continuously. Continuous EEG signal x(t) for TFR $\chi_{\omega }\left(a,b\right)$ can be expressed as:(1)$$\chi_{\omega }\left(a,b\right)=\frac{1}{\sqrt{\left|a\right|}}\int_{-\infty }^{\infty }x\left(t\right)\bar{\psi }\left(\frac{t-b}{a}\right)dt,$$
where, *a* is the scaling parameter that should be greater than 0 and *a*∈*R*; *b* is translation and *b*∈*R*; *t* is the time instant; $\bar{\psi }$(t) is called the mother wavelet which is also continuous for both time and frequency domain.

Mother wavelet provides the scaling and translation of the original wavelet x(t). The original signal is then formulated again using Equation (Equation 2).
(2)$$x\left(t\right)=C_{\psi }^{-1}\int_{-\infty }^{\infty }\int_{-\infty }^{\infty }\chi_{\omega }\left(a,b\right)\frac{1}{\sqrt{\left|a\right|}}\tilde{\psi }\left(\frac{t-b}{a}\right)ab\frac{da}{a^{2}}$$
where $C_{\psi }$ is called the wavelet admissible constant whose value satisfies between

$$0<C_{\psi }<\infty$$

And expressed as:(3)$$C_{\psi }=\int_{-\infty }^{\infty }\frac{\bar{\hat{\psi }}\left(\omega \right)\hat{\tilde{\psi }}\left(\omega \right)}{\left|\omega \right|}d\omega$$

An admissible wavelet must integrate to zero. For this x(t) is recovered by using the second inverse wavelet shown in Equation (Equation 2).

(4)$$x\left(t\right)=\frac{1}{2\pi \bar{\hat{\psi }}\left(1\right)}\int_{-\infty }^{\infty }\int_{-\infty }^{\infty }\frac{1}{a^{2}}\chi_{\omega }\left(a,b\right)exp\left(\frac{i(t-b)}{a}\right)dbda$$

The wavelet for each time is defined as:(5)ψ(t)=ω(t)exp(it)
where, ω(t) is the window. Using the filterbank in CWT, we gather TFR of each channel. Figure 3 represents the TFR of different classes of SEED and DEAP. The TFR of different classes can be differentiated. Before extracting features, TFR images are first resized to 227×227 and separated for testing and training the model using 20x fold validation.

### 3.2. Feature Extraction

In this paper, the time and frequency domain features were extracted using AlexNet. AlexNet performance was superior to earlier methods. AlexNet is a multilayer deep neural network (DNN) model consisting of 23 layers followed by the classification output layer [44]. AlexNet has an extensive network structure of 60 million parameters and 650,000 neurons [45].

AlexNet is a network of vast deep neural networks (DNN) with the combination of three layers: convolutional, max pooling, and fully connected layer. The TFR images were forwarded through the pre-trained AlexNet model. Figure 4 shows the overall architecture of AlexNet used in this article. AlexNet has five convolutional layers, followed by three fully connected layers. We did not use the last segment in our work, which is softmax and performs the classification. The main advantage of AlexNet is its rapid down sampling through stride convolutions and max-pooling layers. The AlexNet model has low computational complexity due to its lower layer count in comparison to other models. The two-dimensional TFR images are first resized to 227×227×3 before extracting features. The images used are the 2D representation of an EEG signal. Therefore, different emotional states have different corresponding regions. Hence, it is more sensitive to differences in spatial information between different states. Each feature obtained from the last fully connected layer has 4096 attributes. The ‘fc7’ layer outputs the feature vector of each channel.

In Figure 4 the TFR images are convolved using all conv layers and max-pooling is used to reduce the dimension of an image after each convolution. Rectified linear unit (ReLU) is used as the activation function and is expressed as

(6)ReLU=max(0,x)

It selects the maximum value between 0 and the input x. The full connection layer ’fc7’ gives the output with 4096 attributes. Figure 5 represents the tree of the output feature dimension vector of both the datasets. SEED dataset uses three classes; the experiment was performed on 15 subjects with five trials of an experiment for each category. The number of channels used for each participant was 62 [33]. After two-dimensional representation, the signal across each channel is represented as an image. Therefore, 62 images are created for each participant using TFR with the 4096 dimension vector of the ‘fc7’ layer. Hence, the total output feature vector obtained from the SEED dataset is 13,950 ×4096. Similarly, for the DEAP dataset, the nine class data were performed on 32 subjects twice generates a feature vector of 18,432 ×4096.

### 3.3. Bag of Deep Features (BoDF)

Bag of features has been used to order less collection of features for image classification in computer vision [46]. We proposed the bag of deep features (BoDF) model for the reduction of features up to suitable value. Using this model, the feature size is significantly reduced to save time for training the dataset. A large number of features usually take long processing time and poor classification performance. In our article, we have been using all the channels to recognize emotions; therefore, feature with high dimension results in high computational cost for training the dataset. Previously researchers used only eight or 12 channels, which also results in lower accuracy by ignoring the rest of the channels. Hence, there is a trade-off between accuracy and the total number of channels used for classification [47].

In this paper, we utilize all the channels of SEED and DEAP dataset and achieve better results. Figure 6 illustrates the overall structure of the proposed method in which the total number of features with dimension 13,950 ×4096 for SEED dataset and 18,432×4096 for DEAP dataset are fed to BoDF model for feature reduction. The BoDF model consists of two steps in which features extracted from AlexNet are reduced using k-means clustering. k-mean clustering groups similar features and represent one feature vector. The first stage reduces the feature vector based on mean clustering. Features are further reduced in the second stage by calculating the histogram features. This model selects the number of features that are closer to the centroid for each class while ignoring others, which are not very useful features for emotions.

#### 3.3.1. Stage 1: k-Mean Clustering

In step 1 of BoDF, the features are extracted from the AlexNet model of each class clustered using k-means clustering. Clustering is used to group similar features that belong to the same class. The k-means algorithm suits well for large datasets while other available clustering techniques suffer overfitting when dealing with large dataset [48]. The k-means algorithm is used to group redundant features by comparing the distance Initially, k is chosen randomly, and k is the number of clusters to group the features. At k = 10, all features are clustered correctly. That is, the features of the two data sets are clustered in 10 comparable groups. First, the distance of each characteristic value is calculated using Euclidean distance. The distance formula is used to compare each feature value and goes to the specific cluster with the shortest distance. For each cluster, the mean value is calculated by taking the average of all attribute values in a specific cluster. The average feature values are then re-evaluated until the average of the centers converges. For SEED, feature vectors of 13,950 × 4096 and 18,432 × 4096 are clustered in 30×4096 and 40×4096 in the SEED and DEAP datasets, respectively. The 10×4096 for each class feature vector is called a vocabulary.

Choosing ’k’ clusters (centroids) is also a difficult task for achieving the best possible results. The universal answer for determining the value of k does not exist and starts with an arbitrary value instead. Hit and trial methods were used to select the value of k. This commonly used method is to measure the difference between the sum of the results of squared errors at different values of k. Sum of the squared errors for all values of k is calculated using Equation (Equation 7).
(7)$$\mathrm{Sum}\;\mathrm{of}\;\mathrm{the}\;\mathrm{squared}\;\mathrm{errors}=\sum_{i=1}^{k}{\left(x_{i}-c_{i}\right)}^{2},$$
where, $x_{i}$ is the input feature vector; $c_{i}$ centroid of the ith cluster; *k* is the cluster number.

In Figure 7 graph shows that the value of the sum of the squared errors is quite small at k = 10.

#### 3.3.2. Stage 2: Histogram Features

The histogram of vocabulary is calculated in the second step. The vocabulary compares each feature vector of 30×4096 and 40×4096 with the EEG data set of all channels. For the SEED dataset, we used features one by one at 30×4096 to compare with 62 channel features and calculate the frequency of occurrence. So we get a histogram feature of 225×30 with each attribute value being a histogram between 0 and 30. Similarly, we get a histogram feature vector of 576×40 for a DEAP dataset with four classes and 32 channels. This step has considerably reduced the feature size. The histogram features indicate the frequency of the best feature for each class.

### 3.4. Classification

As stated earlier, we use the AlexNet model only for extracting features; we did not use the last three layers of AlexNet models, which performs classification of the provided data. AlexNet is a pre-trained model and has the capability of recognizing up to 10,000 objects [44]. In our case, we have three and four classes for SEED and DEAP datasets, respectively. Therefore, our classifier selection is based on the classification output. For this work, we select a support vector machine (SVM) and k-nearest neighbor (k-NN) classifiers that outperform the classification performance based on four classes.

In this study, emotions of both datasets from all the kernels of SVM [49] and k-NN [50] are used for classification purposes. The principle of each classification is based on aggregation. In the SVM classifier [51], data is assigned to higher dimensions the optimum hyperspace for separation of space and data is built in this space. This classifier is a secondary programming problem [51]. In the training phase, SVM creates a model, maps the decision boundaries of each class, and specifies the hyperplane that separates the other classes. Increasing the hyperspace margin increases the distance between classes for better accuracy of the classification. SVM is used because it can effectively perform for non-linear classification [52]. In this study, BoDF feature vector is fed to SVM classifier to distinguish between happy, sad, and healthy emotional states for SEED dataset and two-dimensional Valence and Arousal states for the DEAP dataset. The SVM classifier is the kernel-based classifier. The kernel function is the mapping procedure performed on a training set to improve the correspondence with a linearly separable dataset. The purpose of mapping is to increase the dimensions of the dataset and execute it efficiently using kernel functions. Some of the most commonly used kernel functions are linear, quadratic, and polynomial Gaussian (RBF) kernels.

The second classifier we used for classification is the k-NN classifier. It is the instance-based classifier that classifies objects based on their closest training examples in feature space [52]. The object is classified as a majority of neighbors; that is, the object is assigned to the most common class of k-nearest neighbors, where k is a positive integer [53]. In the k-NN algorithm, the classification of the new test feature vector is determined by the class of the k-nearest neighbors [54]. Here in our work, we implement a k-NN algorithm using a Euclidean distance metric to find the nearest neighbor. The Euclidean distance between two points *x* and *y* is calculated using Equation (Equation 8). Where *N* is the number of features; 45 and 288 for SEED and DEAP dataset, respectively. The classification performance is determined on different values of k ranging from 1–10, and results showed that the best accuracy achieved at k = 2.

(8)$$d\left(x,y\right)=\sum_{i=1}^{N}\sqrt{x_{i}^{2}-y_{i}^{2}}$$

For accessing the classification performance using SVM and k-NN classifiers, we used 20× cross-validation, which is useful for classification [55]. Classification performance achieved from both the classifiers is discussed with the results in the next section.

## 4. Results and Discussion

The accuracy performance of the proposed BoDF-method is evaluated on both the SEED and DEAP datasets. The features extracted from the proposed method are then classified using all the kernels of SVM and k-NN classifier. The results achieved from both the classifiers are also compared with the earlier studies and benchmark results of SEED and DEAP dataset. Feature vector extracted from AlexNet are reduced to 225×30 and 576×40 dimension vector for SEED and DEAP dataset, respectively in BODF-model. After classification, three emotional states positive/negative/neutral of all 15 participants of SEED dataset are classified.

Figure 8 shows the variation in classification accuracy for different kernels of the SVM classifier for two datasets. Linear and cubic classifiers show minimum variation in the classification accuracy as compared to other kernels. The maximum classification accuracy of 93.8% is achieved using the cubic kernel, as evident from the figure. Figure 8b shows the classification accuracy with the k-NN classifier. The result shows that the fine kernel of k-NN works better for CWT based features. Figure 8b shows the variation in classification accuracy for different kernels of the k-NN classifier. Cosine and coarse classifiers show minimum variation but minimum classification accuracy as compared to other kernels. The maximum classification accuracy of 91.4% is achieved using the fine and medium kernel, as evident from the figure.

The average classification accuracy on both datasets using SVM and K-NN classifiers are shown in Table 3.

In reference to the comparison table, our proposed model has a higher classification accuracy than earlier studies. It can be seen that the classification accuracy achieved from the SVM classifier is more as compared emotion recognition using the k-NN classifier (see Table 4). BoDF model proposed in this paper recognize emotions without channel decomposition.

The proposed BoDF model employs the vocabulary set of size 30 features, which produces a small feature dimension. However, the lowest computation complexity is possible by using the AlexNet model with the lowest number of layers. Feature reduction based on BoDF model also shows that using clustering and selecting mean features results in higher classification accuracy. Figure 9 clearly indicates that higher efficiency is achieved when trials or channels are reduced to its maximum value.

The lower feature dimensionality provides the superiority of the proposed architecture on other models in terms of its memory requirement. Secondly, the proposed model can more accurately classify the emotion based on the EEG signal in comparison to other models. However, the proposed model requires the vocabulary set before the feature extraction process. This will increase the computational complexity of the model. In the future, more robustness in recognizing emotion with low computational complexity can be achieved using different pooling algorithms.

## 5. Conclusions

This emotion recognition model, based on bag of deep features (BoDF), is intended to achieve a higher classification accuracy for SEED and DEAP data sets. The results obtained with the proposed method without channel decomposition are higher than the benchmark results of both datasets. We have also found that the proposed method has better classification performance when the feature size is reduced to the optimum level. Reducing the number of channels also degrades the classification performance. All EEG signals are analyzed based on temporal and spatial characteristics. In the BoDF model, the k-means clustering algorithm is used to reduce the size of the feature without disturbing the overall accuracy of the proposed model. Three states for SEED and four emotional states for the DEAP dataset of each participant are recognized using multiple support vector machines (M-SVM) and k-NN classifiers. The results show that our method is superior to previous studies with the same data set. In the BoDF model without channel degradation, it has been demonstrated that the properties obtained from all channels are useful for classifying human emotions using EEG signals. The rating accuracy obtained with multiple SVMs is higher than the rating accuracy of the k-NN classifier. Our study has shown that it achieves higher classification accuracy compared to the emotion recognition method of the conventional channel.

## Figures and Tables

**Figure 1 sensors-19-05218-f001:**
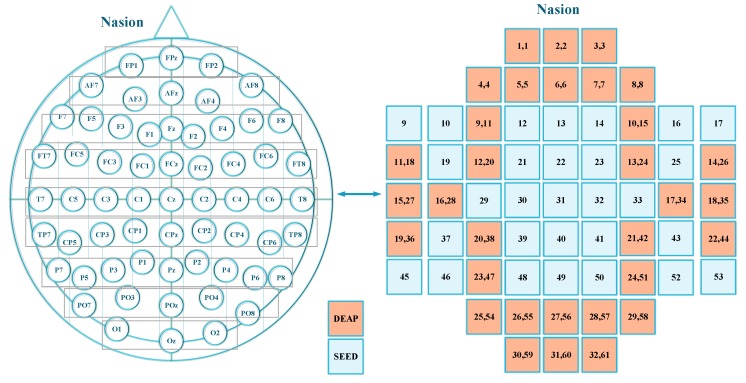
Electrode to channel mapping.

**Figure 2 sensors-19-05218-f002:**
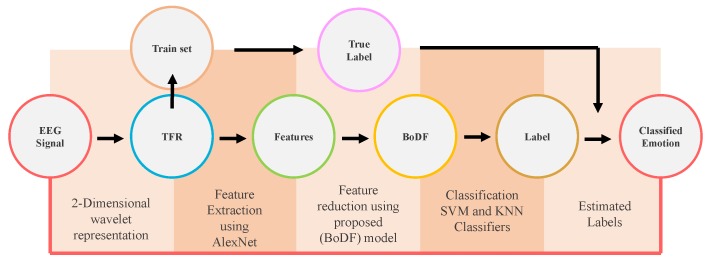
Framework.

**Figure 3 sensors-19-05218-f003:**
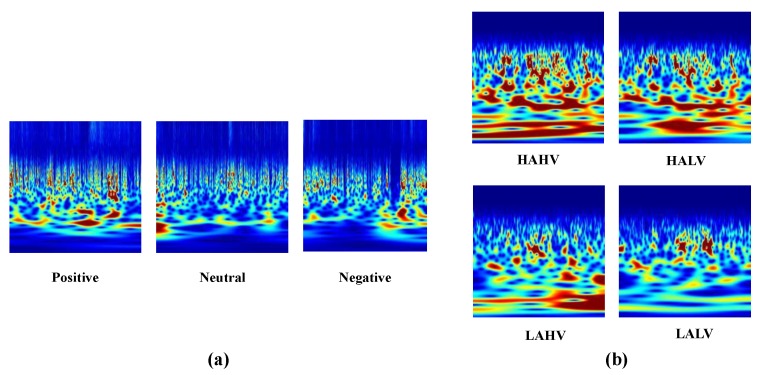
Continuous wavelet transform. (**a**) is the 3 classes TFR of SEED dataset. Positve, negative and netral classes of SEED dataset is clearly differentiatable. (**b**) shows TFR of 4 classes in DEAP dataset.

**Figure 4 sensors-19-05218-f004:**
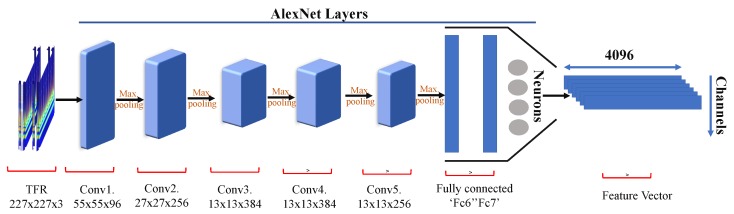
AlexNet layer architecture.

**Figure 5 sensors-19-05218-f005:**
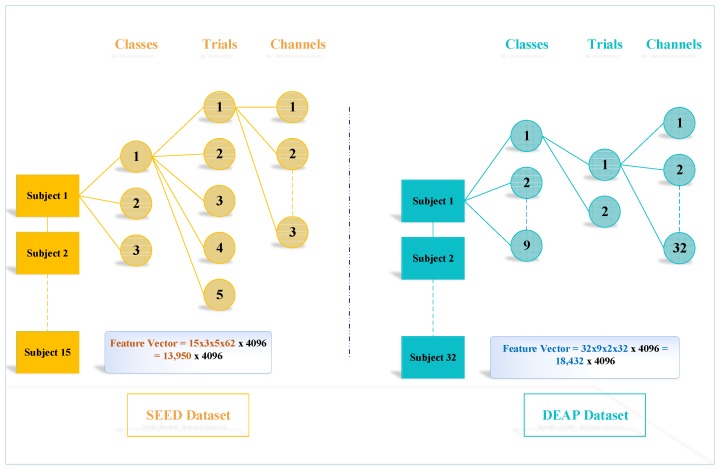
Feature tree.

**Figure 6 sensors-19-05218-f006:**
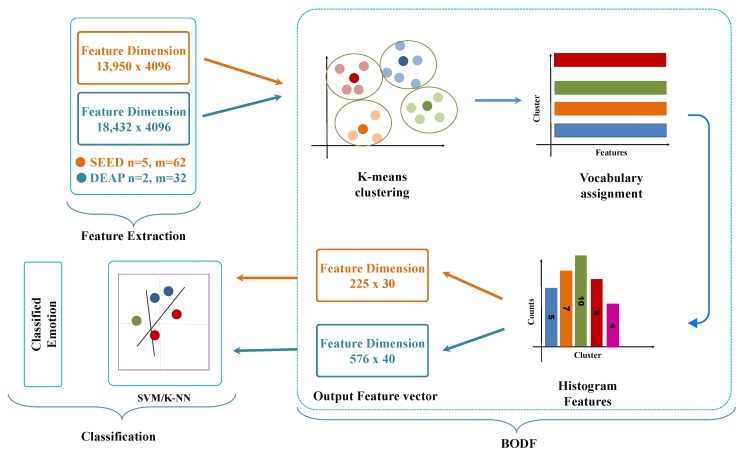
Bag of deep features (BoDF).

**Figure 7 sensors-19-05218-f007:**
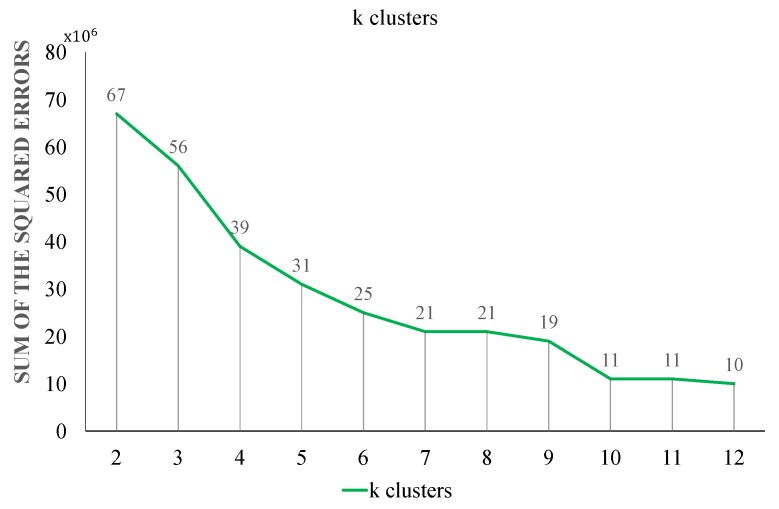
Selecting value of k.

**Figure 8 sensors-19-05218-f008:**
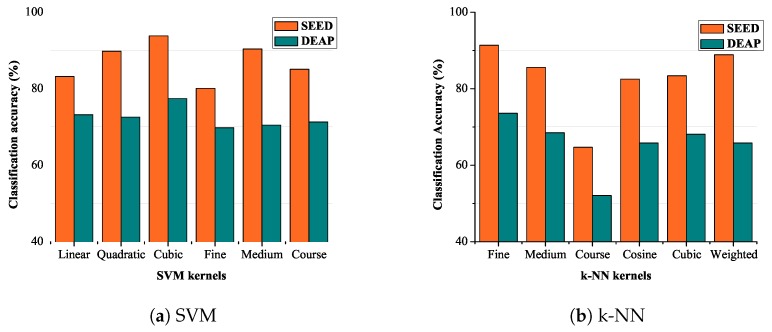
Classification accuracy.

**Figure 9 sensors-19-05218-f009:**
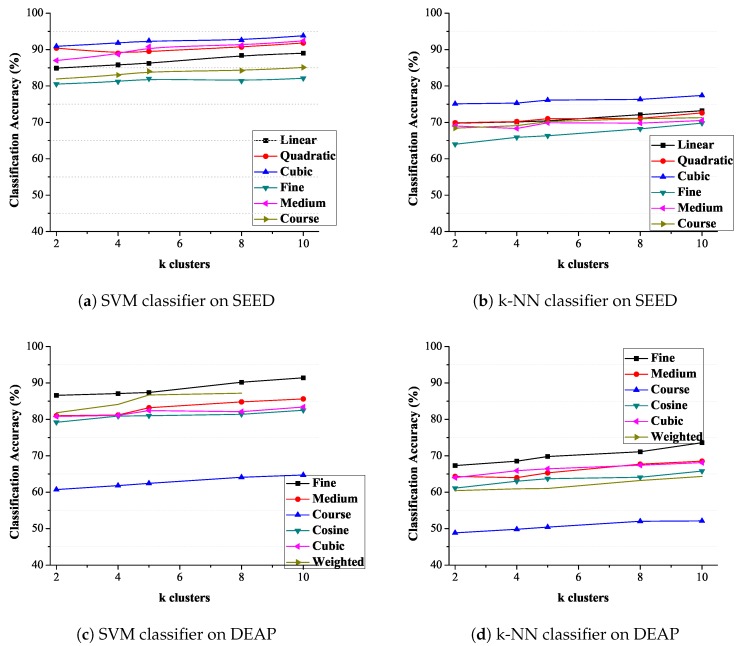
Classification accuracy at different value of k-clusters.

**Table 1 sensors-19-05218-t001:** SEED dataset overview.

No.	Emotion Label	Film Clip Source
1	Negative	Tangshan Earthquake
2	Negative	1942
3	Positive	Lost in Thailand
4	Positive	Flirting scholar
5	Positive	Just another Pandora’s Box
6	Neutral	World Heritage in Chine

**Table 2 sensors-19-05218-t002:** DEAP dataset classes.

No.	Emotion Label	States
1	LAHV (Low Arousal High Valence)	Alert
2	HALV (High Arousal Low Valence)	Calm
3	HAHV (High Arousal High Valence)	Happy
4	LALV (Low Arousal Low Valence)	Sad

**Table 3 sensors-19-05218-t003:** Average classification accuracy (%).

Classifier	SEED		DEAP	
	k Value	Accuracy	k Value	Accuracy
**SVM**	10	93.8	10	77.4
	8	92.6	8	76.3
	6	92.4	6	76.1
	4	91.8	4	75.3
	2	90.9	2	75.1
**k-NN**	10	91.4	10	73.6
	8	90.2	8	71.1
	6	87.4	6	69.8
	4	87.1	4	68.5
	2	86.6	2	67.3

**Table 4 sensors-19-05218-t004:** Comparison on publicly available dataset with previous studies.

Ref.	Features	Dataset	Number of Channels	Classifier	Accuracy (%)
[3]	MOCAP	IMOCAP	62	CNN	71.04
[4]	MFM	DEAP	18	CapsNet	68.2
[17]	MFCC	SEED	12	SVM	83.5
				Random Forest	72.07
		DEAP	6	Random Forest	72.07
[15]	MEMD	DEAP	12	ANN	75
				k-NN	67
[24]	STRNN	SEED	62	CNN	89.5
[26]	RFE	SEED	18	SVM	90.4
		DEAP	12	SVM	60.5
[23]	DE	MAHNOB	18	PNN	77.8
		DEAP	32	PNN	79.3
Our work	DWT-BODF	SEED	62	SVM	93.8
				k-NN	91.4
		DEAP	32	SVM	77.4
				k-NN	73.6

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
