# Peer review of "EEG-Based Multi-Modal Emotion Recognition using Bag of Deep Features: An Optimal Feature Selection Approach"

_sensors, 2019, doi:10.3390/s19235218_

Round 1

Reviewer 1 Report

This paper studies the multi-modal emotion recognition problem and proposes a signal processing method using the deep neural network (DNN) for such an issue based on EEG signals. A temporal based, bag of deep features (BoDF) model is utilized to reduce the features size from all channels. A classification accuracy of 93.8% and 77.4% was achieved for the SEED and DEAP data sets, respectively.

Overall, the paper is well written, and experimental results are clearly presented. However, the performance verification of the proposed method, i.e., the comparisons between the proposed method and those best existing methods, is not enough, and thus, the scientific novelty of the proposed method cannot be fully evaluated. I suggest the authors including more comprehensive results for performance verifications of the proposed method.

Detailed/Major comments:

Section 1. Introduction. The literature review in the introduction is insufficient. Common Spatial Patterns and Signal Decomposition Methods are very popular ways for EEG signal analysis and classification. I recommend to add following articles to improve the literature work of this work, Lotte and C. Guan, “Regularizing common spatial patterns to improve bci designs: unified theory and new algorithms,” IEEE Transactions on biomedical Engineering, vol. 58, no. 2, pp. 355–362, 2010. Song and J. Epps, “Classifying eeg for brain-computer interface: Learning optimal filters for dynamical system features,” Computational intelligence and neuroscience, vol. 2007, p. 3, 2007. Kevric and A. Subasi, “Comparison of signal decomposition methods in classification of eeg signals for motor-imagery bci system,” Biomedical Signal Processing and Control, vol. 31, pp. 398–406, 2017. T. Sadiq, X. Yu, Z. Yuan, Z. Fan, A. U. Rehman, G. Li, and G. Xiao, “Motor imagery eeg signals classification based on mode amplitude and frequency components using empirical wavelet transform,” IEEE Access, vol. 7, pp. 127 678–127 692, 2019. G. Coogan and B. He, “Brain-computer interface control in a virtual reality environment and applications for the internet of things,” IEEE Access, vol. 6, pp. 10 840–10 849, 2018. Section 3. Methodology. Continuous Wavelet Transform (CWT) method reference and mathematical description is missing. Better to give a brief description of such methods before use. Section 3. Methodology. Is there any specific reason to use CWT for comparisons? There are many wavelet-based methods available in literature, e.g. discrete wavelet transform, empirical wavelet transform, empirical mode decomposition etc., for EEG signal classifications. Authors are recommended to add details of such methods in literature and give the reason to use CWT. Section 3. Methodology. There are many feature selection methods available in literature, and no details of any feature selection methods have been included in the literature work. Since one of the main contributions of this work is feature selection, authors are suggested to include brief descriptions of those feature selection methods in literature and draw their potential limitations. Section 3. Methodology. The authors used k-mean clustering as a first step of BoDF. Is there any specific reasons of using k-mean clustering? Why not to use PCA or hierarchical clustering? Please write few lines to justify your claim. Section 3. Methodology. The authors used pre-trained deep convolutional neural network named as Alexnet, which is a very traditional method. There are many other methods e.g. Resnet50, Resnet101, Googlenet, Densenet, Nasnet, Mobilenet etc. The authors need to mention the reasons in comparison with other methods to justify the use of Alexnet. Section 3. Methodology. The authors utilized max pooling layers in their CNN model, which may have many disadvantages. Modern CNN or even ELM use square root pooling or stochastic pooling methods, which are much better than max pooling techniques. Please justify your claim of using max pooling layers in CNN model. Section 4. Results and Discussion. The authors may better discuss both the pros and cons of their method, and briefly describe the future work on this topic.

Minor comments:

The authors used all number of channels, which make the proposed method computationally extensive. It is advice to use channels that provide maximum information related to emotion. The authors can develop automated channel selection criteria or used channels based on the physiological knowledge and compare the results for all and selected channels. The author can draw a graph as number of channels vs. classification accuracy。 It is advised to calculate the computational time of proposed method and suggest whether the method can be used for online application or used for offline applications only. In Fig. 9(b), the label of the x-axis is missing. It is also better to enlarge the font size of the figures.

Author Response

Respected Viewer,

Reviewer 2 Report

Authors tested a features reduction method (Bag of Deep Features) to classify emotions from available datasets. The study is interesting, but some issues have to be fix before its publication. 

Line 21: Authors should make more clear what does it mean "to control devices using brain signals". Are they talking about BCI? In this case, the control feature is not emotion. Please clarify this point.

Line 34: It is not clear the sentence "Each generation has found new ways to understand and discover emotions". Please clarify.

Line 38: Authors speak improperly of BCI and emotion. In fact, in the classical meaning, control feature of a BCI is not the emotion (e.g. refers to https://www.sciencedirect.com/science/article/pii/S093336571300119X). Maybe are they referring to passive BCI? Please clarify. e.g. Refers to https://ieeexplore.ieee.org/abstract/document/7898410 and https://iopscience.iop.org/article/10.1088/1361-6579/aad57e/meta  and https://ieeexplore.ieee.org/abstract/document/8037544 for passive BCI examples.;

Line 39: Authors wrote "in the past", but they should put references.

Line 42: "between them?" Please make it clearer.

Line 54: Authors should report for each study the number of classes that have been used for each study, in order to make the classification accuracies comparable.

Line 67: The sentence is not completed. Please fix the typo

Line 69: Typo (relevant, relevant).

Line 74: From "Experimental results...", it is not clear. Please rephrase the concept.

Line 108: Between 0 and 75 Hz, all the EMG artefacts remain. Also, by not removing 0Hz, there will be fluctuations of EEG signals. Please make a more strict filtering, to avoid any bias effect (e.g. from 1Hz to 30Hz).

Figure 8 should be on the results section.

Line 273: Results paragraph should be separated by Discussion paragraph.

Results: It was not performed any statistical analysis on the presented accuracy. Please address this issue.

In conclusion paragraph, authors should report possible limitation of their study. For example, what about the employment of this technology in real settings? By using all the temporal and spatial features, there would be possible confound factors (e.g. movements, other mental states, etc.)

Finally, a neurophysiological analysis is completely missed. An analysis of chosen features should be performed, to understand what the method is classifying, to avoid any bias effect (e.g. movements).

Author Response

Respected reviewer,

Round 2

Reviewer 1 Report

The authors have successfully addressed all my concerns, and thus, I recommend the publication of this manuscript on Sensors. 

One minor concern: The font size in Figs. 8-9 could be further enlarged. 

Author Response

Comments and Suggestions for Authors

The authors have successfully addressed all my concerns, and thus, I recommend the publication of this manuscript on Sensors. 

One minor concern: The font size in Figs. 8-9 could be further enlarged. 

Reply: The honorable reviewer is thanked wholeheartedly for feedback and improving the quality of the manuscript. As suggested by the reviewer the font size in fig 8 and 9 has been enlarged in the updated document. The manuscript has been updated, and the suggested changes have been incorporated. Thank you once again for the valuable feedback.

Reviewer 2 Report

Authors addressed the most of the highlighted issues, anyhow one comment is still unaddressed. In particular, regarding BCI and passive BCI definition. Passve BCI is not a brain signal acquisition technology, and BCI is not a technique to classify. Machine learning for example could be defined a technique to classify, and bioamplifiers are technology for brain signal acquisition, not BCIs. Authors seem to be confused in this regard. Anyhow they did not refer to the suggested references in this regard. Please address this issue.

Author Response

Comments and Suggestions for Authors

Authors addressed the most of the highlighted issues, anyhow one comment is still unaddressed. In particular, regarding BCI and passive BCI definition. Passve BCI is not a brain signal acquisition technology, and BCI is not a technique to classify. Machine learning for example could be defined a technique to classify, and bioamplifiers are technology for brain signal acquisition, not BCIs. Authors seem to be confused in this regard. Anyhow they did not refer to the suggested references in this regard. Please address this issue.

Reply: The honorable reviewer is thanked wholeheartedly for feedback and improving the quality of the manuscript. Regarding BCI definition, Brain-computer interface or BCI is a technology that sends and receives signals between the brain and an external device. While the process involves in acquisition and classification are BCI components. The simplified BCI definition can describe the technology as a "direct communication link between the brain and external devices", a bidirectional link (bidirectional interface). One direction is for the BCI to send brain activity to the computer and the computer to translate brain activity into motor instructions. Computers can also communicate in other ways and send information directly to the brains of BCI users. This is called active BCI with a direct brain connection compared to non-invasive passive BCI. Non-Invasive BCI is a passive BCI that receives a signal from brain (Passive BCI). Invasive BCI projects signals to the brain (Active BCI). BCI acquires signals through EEG electrodes.

The following reference the author state that BCI measures brain signals, extracts features and translates to output signals to control several devices.

Nijboer, Femke & Carmien, Stefan & Leon, Enrique & Morin, Fabrice & Koene, Randal & Hoffmann, Ulrich. (2009). Affective brain-computer interfaces: Psychophysiological markers of emotion in healthy persons and in persons with amyotrophic lateral sclerosis. Affective Computing and Intelligent Interaction and Workshops, 2009. ACII 2009. 3rd International Conference on. 1 - 11. 10.1109/ACII.2009.5349479.

The esteemed reviewer is thanked wholeheartedly for the valuable point and feedback. The suggested references have been incorporated in the updated manuscript. Thank you once again for clearing out the point regarding BCI definitions